# Histological and Microscopic Analysis of Fats in Heart, Liver Tissue, and Blood Parameters in Experimental Mice

**DOI:** 10.3390/genes14020515

**Published:** 2023-02-17

**Authors:** Sehrish Basheer, Imran Riaz Malik, Fazli Rabbi Awan, Kalsoom Sughra, Sadia Roshan, Adila Khalil, Muhammad Javed Iqbal, Zahida Parveen

**Affiliations:** 1Department of Biotechnology, University of Sargodha, Sargodha 40100, Pakistan; 2Diabetes and Cardio-Metabolic Disorders Lab, Health Biotechnology Division, National Institute for Biotechnology and Genetic Engineering (NIBGE), Jhang Road, Faisalabad 38000, Pakistan; 3Department of Biochemistry and Biotechnology, University of Gujrat, Gujrat 50700, Pakistan; 4Department of Zoology, University of Gujrat, Gujrat 50700, Pakistan; 5Department of Biochemistry, Abdul Wali Khan University, Mardan 23200, Pakistan

**Keywords:** dietary fat, vegetable/plant oil, banaspati ghee, desi ghee, mouse

## Abstract

The intake of various types and amounts of dietary fats influences metabolic and cardiovascular health. Hence, this study evaluated the impact of routinely consumed Pakistani dietary fats on their cardiometabolic impact. For this, we made four groups of mice, each comprising 5 animals: (1) C-ND: Control mice on a normal diet, (2) HFD-DG: High-fat diet mice on a normal diet plus 10% (*w*/*w*) desi ghee, (3) HFD-O: Mice on normal diet plus 10% (*w*/*w*) plant oil (4) HFD-BG: Mice on normal diet plus 10% (*w*/*w*) banaspati ghee. Mice were fed for 16 weeks, and blood, liver, and heart samples were collected for biochemical, histological, and electron microscopic analysis. The physical factors indicated that mice fed on HFD gained more body weight than the C-ND group. Blood parameters do not show significant differences, but overall, the glucose and cholesterol concentrations were raised in the mice fed with a fat-rich diet, with the highest concentrations in the HFD-BG group. The mice fed with HFD-BG and HFD-O had more lipid droplets in the liver, compared to HFD-DG and C-ND.

## 1. Introduction

Cardiovascular diseases (CVDs) upset the heart and blood vessels. The buildup of fats in the inner walls of arteries is associated with the development of CVD. The narrowing of arteries hinders blood flow, damaging vital organs, especially the heart and brain. Cardiovascular diseases mainly include coronary artery disease (CAD), stroke, peripheral arterial disease, and aortic disease [1].

CVD afflicts people of all ethnicities, gender, ages, and socioeconomic levels. CVDs are the leading cause of death worldwide, accounting for more than 32% of total annual deaths in 2019, of which 85% were due to stroke and heart attack. CVD cases have increased substantially from 271 million in 1990 to 523 million in 2019. A surge in CVD-related deaths also reached 18.6 million in 2019, compared to 12.1 million in 1990 [2]. In contrast to other regions, the incidence of CVD is higher in the South Asian population (60% of the total world’s heart disease patients). Moreover, they also develop CVDs at a young age, causing increased premature mortality in South Asia [3]. Overall, 27% of the total deaths are caused by CVDs in this region.

According to the WHO report, Pakistan has the highest prevalence of cardiovascular diseases globally, with 19% of all deaths caused due to CVDs. One in four middle-aged adults has a CVD in Pakistan [4]. The CVD-related deaths in Pakistan have reached about 250,000 per year [5]. At 40 years or older, Pakistanis have 26.9% more CVDs than other South Asians [6].

The likelihood of developing cardiovascular diseases depends on risk factors such as hypertension, diabetes, an unhealthy diet rich in fats (cholesterol, triglycerides), fatty liver disease, tobacco use, physical inactivity, and obesity. Obesity is an independent risk factor for developing cardiovascular diseases, leading to excessive fat storage in cells, such as adipocytes, hepatocytes, and muscle cells [7]. The higher amounts of lipids can lead to chronic inflammation in the blood vessels, which interferes with the smooth flow of blood, causing coronary artery disease, atherosclerosis, hypertension, and ultimately, heart failure [8].

A significant factor that has tremendously influenced cardiovascular health is poor choice and higher intake of dietary fats. Different investigations show that trans fatty acids (TFA) positively associate with CVD development [9]. The consumption of higher amounts of fats affects the serum lipid profile, increasing low-density lipoprotein cholesterol (LDL-c) and decreasing high-density lipoprotein cholesterol (HDL-c) [10].

A diet enriched in saturated fatty acids (SFA) coupled with physical inactivity induces the buildup of fats and cholesterol in the arteries, which causes atherosclerosis that results in stroke, heart attack, and myocardial infarction (MI). Atherosclerosis is a complex multifactorial pathological condition characterized by fatty plaque deposition in the coronary arteries (arteries that supply blood to the heart tissues) [11]. Cholesterol is a major component of these arterial plaques. The formation and progression of plaque is a complex process, involving various morphological, biochemical, and cellular changes in the walls of blood vessels [12]. Studies have reported that high levels of low-density lipoproteins (LDL) cholesterol and apolipoprotein B-100 (ApoB-100) (the structural protein of LDL) as the main pathological initiators for plaque formation [2].

Moreover, consuming a lipid-rich diet can be associated with excessively adding lipid droplets in the hepatocytes. The liver regulates lipid metabolism by incorporating lipids in bile and producing very low-density lipoproteins (VLDL) [13]. However, dyslipidemia in the form of high levels of serum triglycerides, LDL-C, low levels of HDL, and overproduction of VLDL, accompanied by the blockage in their secretion, poses the threat of a buildup of fats in the liver, which prompts the condition of non-alcoholic fatty liver disease (NAFLD) [14]. Many studies have highlighted the association of NAFLD with hypertension and coronary artery diseases, which are the main cause of CVD morbidity [15,16].

The dietary patterns differ with cultures worldwide and so does the risk of CVD in different populations. South Asian culture is enriched with cuisines that require fats (vegetable/plant-based oil, hydrogenated oil/ banaspati ghee, milk-derived fat/desi ghee, or butter) to be cooked and fried, making the consumption of fats higher than in western countries.

Butter and desi ghee (homemade butter when heated up to remove water content) are saturated fats that are assumed to be safe, compared to industrially prepared cooking oil (unsaturated fats) or banaspati ghee (derived from the hydrogenation of plant-based saturated oils). Traditionally desi ghee has been considered a pure and beneficial source of fat, due to its direct production from milk, most often at home. Although expensive, desi ghee is highly used in rural areas of Pakistan, due to its perceived beneficial effects on health. No scientific study has been conducted before to determine the impact of desi ghee on human health, and very limited data is available on the effects of such diets at the tissue level.

Henceforth, we conducted the present study in laboratory mice to investigate the effects of home-prepared, dairy-based desi ghee, industrial cooking oil, and banaspati ghee on cardiovascular health. Four groups of mice were fed with different fats for 16 weeks, and the effect of these fats on weight, serum lipid profile, and heart and liver histology were investigated.

## 2. Materials and Methods

### 2.1. Preparation of Animals for Experiments

The study was conducted in Diabetes and Cardio-Metabolic Disorders laboratory at the National Institute for Biotechnology and Genetic Engineering (NIBGE) in Faisalabad, Pakistan. The institutional review committee approved the study for animal experimentation.

In this diet (10% *w*/*w* high fat diet, HFD) intervention study, female BALB/c mice were used, with 20 mice divided into 4 groups of 5 animals per group. All mice were kept in the NIBGE animal facility on 12 h light and dark cycles with the supply of food and ad libitum water and ambient temperature. Food was normal chow for control mice or with the addition of 10% (*w*/*w*) of a specific type of fat for the experimental group, as mentioned below.

C-ND: Control mice on a normal diet.HFD-DG: Mice on a normal diet with 10% (*w*/*w*) desi ghee (dairy-derived natural fats).HFD-O: Mice on normal diet with 10% (*w*/*w*) vegetable/plant-based oil.HFD-BG: Mice on a normal diet with 10% (*w*/*w*) banaspati ghee (hydrogenated vegetable oil).

The body weights of all mice were recorded weekly. After 16 weeks of dietary intervention, mice fasted for 16 h before taking their blood samples for biochemical analysis. Animals were then euthanized, organs (i.e., heart, abdominal aorta, and liver) were collected for further analysis, and each organ was weighed.

For histology (light microscopy/transmission electron microscopy), whole heart and part of the liver were fixed in formalin. Electron microscopy processed the abdominal aorta and part of the liver for histological changes to be viewed.

### 2.2. Biochemical Analysis

Clinically significant biochemical parameters, such as blood glucose, lipid profile (cholesterol, HDL-cholesterol, LDL-cholesterol, and triglycerides), alanine aminotransferase (ALT), aspartate aminotransferase (AST), total protein, albumin, creatinine, and uric acid, were measured for serum sample from each animal using commercially available kits on a semi-automated clinical chemistry analyser (Microlab300).

### 2.3. Histological Analysis

Heart and liver tissues were processed for histological analysis. Samples were fixed in formalin. These paraffin-embedded tissues were cut into thin sections and analyzed under the microscope at 40× magnifications after staining with haematoxylin and eosin (H&E) stain. Image J software was used for the image analysis.

### 2.4. Microscopic Analysis

Microscopic analysis of the mouse abdominal aorta and liver was performed. Tissues were dipped in 5% glutaraldehyde prepared in 0.2 M phosphate buffer at pH 6.8. A total of 1% osmium tetroxide was used for post-fixation at room temperature for 18 h; 15 min washing was performed with sterilized distilled water and then dehydrated with ethanol diluted at 30%, 50%, 70%, and 100%, as the ethanol does not give the proper fusion with spur resin, which is why the tissue was left in absolute acetone twice at 30 min as a transitional solvent. The ratio of resin-to-acetone was as follows 1:3 for 18 h, followed by 1:1 and 3:1 for 18 h each. The final 100% resin mixture was added to the samples and left overnight. The samples were poured into moulds and incubated at 60 °C for 48 h.

The polymerized resin blocks were cropped and tackled with a fine scalpel blade before thin or semi-thin serial sections of 5 microns were cut with Ultra Microtome (RMC MT 7000). Staining was performed with 5% toluidine blue prepared in 1% *w*/*v* boric acid (H_3_BO_3_) for 10 min at 60 °C. Glass slide-mounted sections were examined with Nikon Light Microscope.

Transmission electron microscopy, ultra-thin serial tissue sections of approximately 120 nm were placed on 300 mesh nickel grids and stained the grids with 5% uranyl acetate for 30 min. Double-stained the grids with lead citrate for 10 min in a NaOH chamber. Examine the grids under transmission electron microscope (JEOL JEM 1010) at 80 Kv.

### 2.5. Statistical Analysis

Continuous variables were expressed as mean ± standard deviation. Percentage increase or decrease was calculated by comparing the values of experimental groups (HFD-O, HFD-DG, and HFD-BG) with the control group.

## 3. Results

### 3.1. Types of Dietary Fat and Weight Gain in Mice

The impact of three dietary fats was assessed on the body weight and biochemical parameters of experimental mice. While monitoring the body weight, it was observed that mice fed with HFD-O and HFD-BG gained substantial weight, relative to mice given HFD-DG and a normal diet. Mice fed with the HFD-O group gained maximum weight (Figure 1). A significant shift in weight gain in the HFD-O and HFD-BG from HFD-DG and C-ND mice was observed in the 8th week and onwards (Figure 1). However, these bodyweight changes were not significantly different statistically.

### 3.2. Effect of Different Dietary Fats on Blood Biochemical Parameters

The analysis of biochemical parameters in C-ND, HFD-O, HFD-DG, and HFD-BG mice showed that blood glucose levels were ≥30% higher in groups fed with a high-fat diet (HFD-O = 30%, HFD-DG = 35%, and HFD-BG = 38%), as compared to control mice fed on a normal diet (Table 1).

Analysis of the lipid profile demonstrates that a high-fat diet caused an imbalance in the concentrations of specific lipid parameters. HFD-O and HFD-BG showed a 21% and 22% increase in total cholesterol, compared to the C-ND group. However, in HFD-DG, the increase was only 4%, much less than the other fat-fed groups (HFD-O and HFD-BG). This can be linked to the fact that HFD-O and HFD-BG are prepared industrially with added preservatives. At the same time, HFD-DG is naturally derived from milk cream (i.e., homemade), without any further additives or preservatives.

Similarly, the reduction in HDL-C (considered good cholesterol) concentration was nearly two-fold more in HFD-BG and HFD-O mice at 15% each, relative to HFD-DG-fed mice (at 8%). Moreover, there was a 4% increase in LDL-C in HFD-O and HFD-DG-fed mice, while an 11% increase was observed in HFD-BG-fed mice, compared to C-ND mice (Table 1).

Overall, the serum uric acid concentration was higher in mice fed a fat-rich diet. A substantial augmentation was witnessed in the HFD-BG group, which was 85% more than in C-ND. The oil group showed a 41% variation, and HFD-DG showed a 56% change, a moderate variation from HFD-O and HFD-BG (Table 1).

### 3.3. Histological Analysis

Heart and liver samples from control and high-fat diet mice were stained with hematoxylin and eosin. Results indicated no gross morphological change after 16 weeks in the heart (Figure 2)—still, lipid droplets (L.D.s) accumulated in the liver (Figure 3). There were no L.D.s found in C-ND, but little congestion of cells was observed.

However, L.D.s deposition was more in HFD-BG and HFD-O, relative to HFD-DG. All experimental mice, except C-ND, also observed apparent degeneration of cells, necrosis, and infiltration of inflammatory cells.

### 3.4. Transmission Electron Microscopy

Liver sections were processed for imaging by transmission electron microscopy to see any pathological changes in the cellular architecture. Accumulation of lipids in the C-ND group was found to be normal. At the same time, there was a considerable accumulation of lipid droplets in the mice on the HFD-O and HFD-BG diet, compared to the HFD-DG group mice (Figure 4). 

## 4. Discussion

The current study is the first report from Pakistan assessing the impact of commonly used dietary fats (vegetable oil, banaspati ghee, and desi ghee) on cardiac health in the mouse model. Results of the biochemical analysis showed that the mice fed with oil and banaspati ghee gained more weight and showed a deranged lipid profile, compared to the controls. Uric acid was also higher in HFD-O and HFD-BG groups. Although the concentration of lipid profile, including total cholesterol, triglycerides, and LDL, was higher in the HFD-DG group than in control, this increase was less than in the other high-fat diet groups (HFD-O and HFD-BG). Histological analysis of liver sections also complemented these findings and showed a higher accumulation of lipid droplets in the HFD-O and HFD-BG groups. Although the amount of fat fed to the experimental mice was the same in all groups (10% high-fat diet and normal diet), the source of fat and its composition were different. Fats are the major component of the diet in our population, and the common sources are: (1) dairy (butter or desi ghee), (2) plant oils (sunflower oil, canola oil, etc.), and (3) banaspati ghee (obtained from plant oils after hydrogenation). 

Desi ghee (clarified butter or liquid butter) is more specifically used in Asian countries, especially Pakistan and India. Studies from these regions also complement the current study’s findings by showing that desi ghee helps reduce total cholesterol and LDL-C. Moreover, desi ghee helps attenuate LDL oxidation [17] and protects from atherosclerosis and other cardiovascular diseases. A prospective study of 521,120 individuals concluded that substituting vegetable oils for butter and margarine can lower the risk of all-cause mortality or CVD-related mortality [18]. While counting the beneficial effects of oils, their composition and heating cycles should also be considered. Some oils, such as palm oil and coconut oil, contain more saturated fatty acids (SFA) [19] than sunflower oil, canola oil, or olive oil. Thus, the quality of the oil is equally important in assessing its beneficial effects on cardiovascular diseases. Intercomparison of different oils showed that polyunsaturated non-dehydrogenated oils are more beneficial than palm oil in reducing the risk of atherosclerosis and myocardial infarction [20].

Furthermore, repeated heating of oils deteriorate its beneficial effects by lipid oxidation and increases the chances of hypertension, vascular inflammation, and atherosclerosis [21]. Chaturvedi et al. compared the effects of heating on the beneficial roles of desi ghee and sunflower oil. They concluded that desi ghee performs better and reduces oxidative stress after passing through a heating cycle of potato frying [22]. This further strengthens the hypothesis that desi ghee is better than oil and banaspati ghee, which is also evident from current study. Furthermore, a study from Pakistan also reported that the majority of the banaspati ghee brands available in Pakistan do not meet the quality standards set by Pakistan Standard and Quality Control Authority [23], which could be a reason of their detrimental effects of human health. Studies have consensus that banaspati ghee (hydrogenated oil) contains higher amounts of SFA and trans unsaturated fatty acids, which make them a potential inducer of cardiovascular diseases. On the basis of above-mentioned studies and results of current study, it can be said that desi ghee is superior in cardioprotective effects than other dietary fats.

Gas chromatography (G.C.) coupled with a mass spectrometer detector showed that commonly used oil brands in Pakistan are rich in polyunsaturated fatty acids (PUFA). At the same time, banaspati ghee contains more saturated fatty acids and trans-unsaturated fatty acids. The percentage of monounsaturated fatty acid was almost similar in oil and banaspati ghee [24]. Desi ghee is an anhydrous butter fat (a.k.a. clarified butter, liquid butter) containing various fatty acids. In Pakistan, commonly used desi ghee is prepared at home by heating butter on a low flame. The G.C. analysis of various commercial and homemade desi ghee exhibited that homemade desi ghee (which is commonly used in our society) contains 73% of saturated fatty acids (SFA) [25], which is even higher than the amount of SFA detected in banaspati ghee [24]. SFA is known to increase the risk of cardiovascular diseases, while PUFA reduces this risk. Several researches have emphasized reducing the SFA intake and increasing the intake of PUFA. Fatty acid constitutes that SFA and PUFA are equally important in determining their effect on cardiovascular diseases. It is observed that medium chain length SFA is better than long chain SFA, in terms of its effect on lipogenesis and haptic fat deposition [26]. 

In our society, where fat is considered an essential component of diet and foods are prepared with high amounts of fats, it is a common perspective that desi ghee is best among all other fat types because it is minimally processed and made at home without using any industrial procedures, while vegetable oil and banaspati ghee are prepared by industries by chemical processing of seeds or partial hydrogenation of vegetable oils and are, thus, considered less healthy. Research has shown that oil is superior to banaspati ghee, in terms of its health benefits, as it has less SFA and is rich in PUFA. However, the qualitative analysis of desi ghee, oil, and banaspati ghee by Raman spectroscopy showed the presence of distinctive isomers of conjugated linoleic acid in desi ghee, which was missing in banaspati ghee and oil [27]. Scientists have admired conjugated linoleic acid for its anti-oxidant and anti-atherogenic properties [28]. Thus, the presence of conjugated linoleic acid isomers gives desi ghee distinctive properties, making it superior to other dietary fats.

Moreover, desi ghee also contains phospholipids, especially cephalin, which act as an anti-oxidant, and the presence of short (12%) and medium chain fatty acids (20%) helps in its direct absorption to the liver and sustained supply of energy via β-oxidation and krebs cycle [25]. The structure and length of saturated fatty acids also impact their function. Medium chain length SFA reduces fat deposition [26], which subsequently prevents obesity and increases insulin sensitivity. This could be the main reason for the low body weight in our experimental mice fed with desi ghee. The general perception of our Asian society and scientific evidence converge to the conclusion that a limited intake of desi ghee can reduce the risk of cardiovascular diseases. A cohort study on the Iranian population also complemented this conclusion by showing a significant decrease in the atherogenic plasma index in butter and oil users [29]. Furthermore, desi ghee also possesses other health benefits, such as increasing physical and intellectual stamina [30], sustained energy supply, and strengthened immune system [31].

One of the major risk factors for CVD consequences is unhealthy dietary intake. The Pakistani diet has a predilection towards foods rich in carbohydrates and fats. Traditional cuisines are also cooked and fried in rich oil or Banaspati ghee containing trans-fatty acids [32]. In addition to hydrogenated vegetable oil (Banaspati ghee), butter and desi ghee (heated butter fat) are consumed in cooking. A variety of banaspati ghee available in the Pakistani market shows that not all are safe for cooking, due to their substandard quality. Though cheaper and readily available, these cooking oils are thought to cause detrimental effects on the heart, liver, and kidneys [33]. This high-fat diet and physical inactivity give rise to potential risk factors for the development of CVD.

The current study was undertaken to evaluate the outcome of a high-fat diet, natural (desi ghee), and industrial sources (vegetable banaspati ghee and oil) in the mouse model by investigating blood chemistry and vital organs (e.g., heart and liver). Pronounced impacts of different dietary fats were observed on body weight, blood sugar, cholesterol, triglycerides, LDL-C, and HDL. 

Among diet intervention groups, HFD-oil and HFD-banaspati ghee were seen to increase mice’s body weight substantially, and their long-term use can lead to obesity. Obesity-driven metabolic syndromes are linked with the development of type 2 diabetes, established on insulin resistance and cardiovascular disease [34,35].

A Chinese study has reported a correlation between obesity and cardiovascular morbidities. The study demonstrates a high risk of cardiovascular morbidities with obesity, corroborating obesity as an independent risk factor of cardiovascular diseases [36]. A similar study reported an increase in carotid artery intima-media thickness in asymptomatic overweight or obese people [37], which can cause narrowing of the carotid artery and lead to stroke in the future. However, our study found weight gain in mice groups fed with C-ND and HFD-DG within the normal range. This suggests that desi ghee has no significant impact on weight gain and the onset of obesity. 

Serum cholesterol levels were 21% HFD-O and 22% HFD-BG, compared to HFD-DG (4%) and C-ND (7%). This high serum cholesterol, due to HFD-O and HFD-BG, can cause hypercholesterolemia, which causes the onset of obesity, insulin resistance, and decreased HDL levels. Dietary cholesterol is associated with reducing the fatty acid oxidation that increases hepatic and plasma triglyceride levels. Desi ghee can be a safer substitute, as the serum cholesterol level was nearly equal to the C-ND group. Studies have reported the association of the reduction in total cholesterol level with the reduced risk of atherosclerosis [38]. 

Serum LDL-C levels were increased in all high-fat diet groups. In contrast, BG increase was 11%, compared to a 4% increase in groups fed with oil and DG. Our results are consistent with other studies [39], which described that butter intake was associated with increased levels of LDL-c, compared to extra virgin olive oil or coconut oil. Contrary to LDL-C, our study found HDL-C levels to be decreased; HFD-O- and HFD-BG-fed groups manifested a reduction of 11% and 8% in HFD-DG. 

Histological analysis found no significant impact of a high-fat diet on heart tissues. This may be because the trial was conducted for only 16 weeks, which is not sufficiently long for the BALB/C mice to develop a fat reserve in heart tissues. However, lipid deposits were seen in the tissues of the liver. 

Electron microscopic analysis of hepatocytes revealed lipid droplet accumulation in all high-fat diet groups. L.D.s in the liver act as fat reserves, while more than 5% fat of liver weight is considered to be associated with non-alcoholic fatty liver disease (NAFLD) [40]. Groups fed with HFD-O and HFD-BG developed large lipid droplets (L.D.s), compared to a group of mice given DG. A large number of lipid droplets accumulated in hepatocytes can lead to fatty liver, which, in the future, may silently contribute towards the development of heart disease due to dyslipidemia. Fatty liver can cause diseases such as fibrosis, cirrhosis, and hepatocellular carcinoma. It is also frequently associated with obesity, diabetes, dyslipidemia, insulin resistance, and metabolic syndrome. 

This trial comprised a brief period of 16 weeks, yet a considerable amount of lipid droplets were observed in the oil- and BG-fed groups. Thus, it can be inferred that prolonged intake of industrially sourced fat, both HFD-O and HFD-BG, can cause a buildup of fats in the liver. Evidence suggests a strong relationship between fatty liver with the onset of diabetes, cardiovascular diseases, and metabolic disorders [41,42]. A 6-year follow-up study has identified a bidirectional relationship between fatty liver and risk of CVD development [43]. Other studies also reported that fatty liver is related to the increased threat of coronary heart disease and causes abnormality in myocardial structure and function [44,45].

Overall, we identified substantial differences in weight gain, serum cholesterol levels, and formation of lipid droplets in hepatocytes among high-fat diet groups. In contrast, the group fed with DG was observed to have lower levels than HFD-O and HFD-BG. No significant reduction in LDL-C levels in the mice fed with DG HDL-C was observed to decrease, which is an unexpected incidence, as a similar study conducted with butter intake found an increase in HDL-C levels [46]. Furthermore, as stipulated by most villagers taking desi ghee in their diet, it should have increased good fat (HDL-C), which was not found in this study. These findings warrant further research to look for deeper insight. 

## 5. Conclusions

Based on these observations, we conclude that the long-term usage of desi ghee may pose less risk of obesity and obesity-related cardiovascular diseases in the animal models. Histological analysis of the hepatocytes indicates that desi ghee is a relatively safer option than industrially manufactured or hydrogenated oils. Studies with longer trials are recommended to precisely evaluate the effect of these fat sources on the progression of cardiovascular disorders.

## Figures and Tables

**Figure 1 genes-14-00515-f001:**
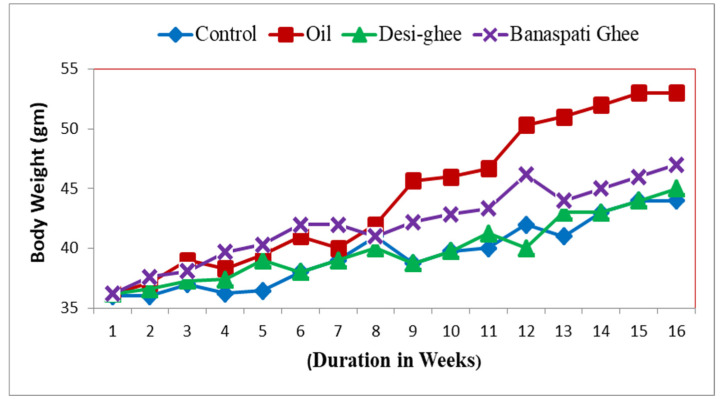
Impact of different dietary fats on body weight in mice.

**Figure 2 genes-14-00515-f002:**
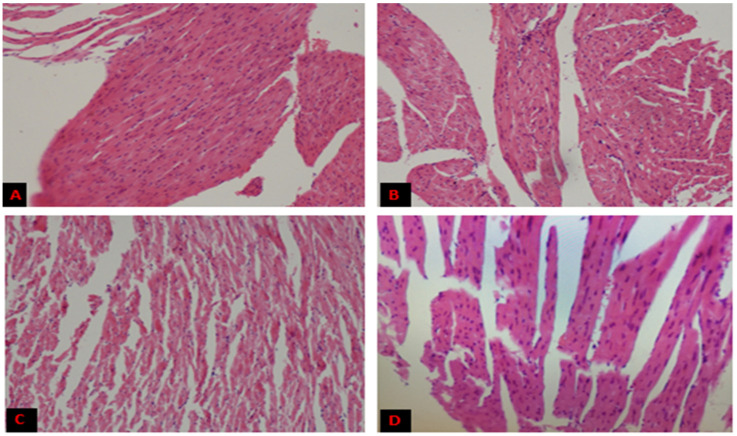
Histopathology of heart sections of mice. Representative sections from mice hearts from C-ND (**A**), HFD-DG (**B**), HFD-O (**C**), and HFD-BG (**D**).

**Figure 3 genes-14-00515-f003:**
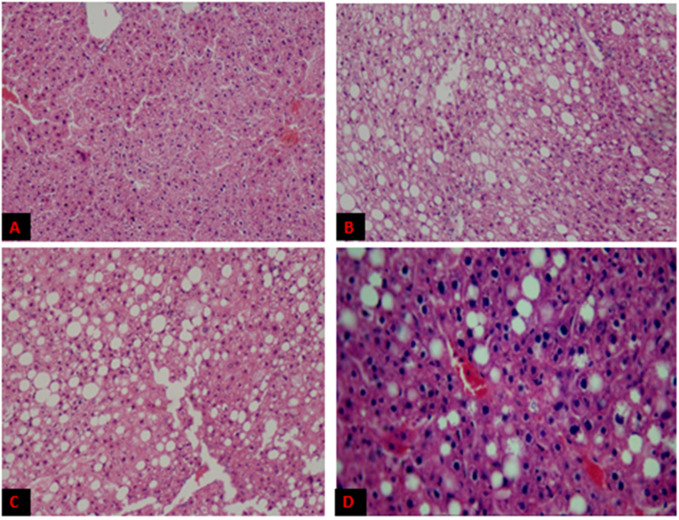
Histopathology of liver sections from experimental mice with fat accumulation depicted as white droplets. Representative sections from mouse liver from C-ND (**A**), HFD-DG (**B**), HFD-O (**C**), and HFD-BG (**D**).

**Figure 4 genes-14-00515-f004:**
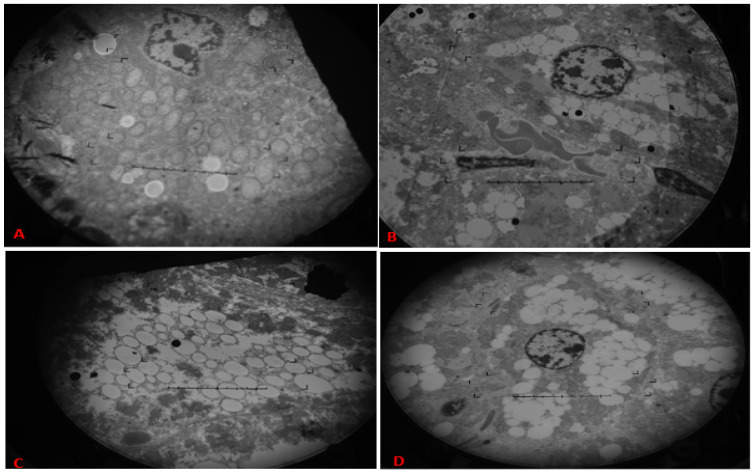
Transmission electron microscopy of liver sections from experimental mice showing lipid accumulation as white droplets. Representative sections from mouse liver from C-ND (**A**), HFD-DG (**B**), HFD-O (**C**), and HFD-BG (**D**).

**Table 1 genes-14-00515-t001:** Effect of fat-rich diet on serum biochemical parameters in mice.

Biochemical Parameters	Mouse Groups			
	C-ND	HFD-O	HFD-DG	HFD-BG
Glucose (mg/dL)	137 ± 3	178 ± 5 (↑30%)	185 ± 3 (↑35%)	189 ± 3 (↑38%)
Cholesterol (mg/dL)	214 ± 7	259 ± 5 (↑21%)	222 ± 3 (↑4%)	262 ± 2 (↑22%)
Triglycerides (mg/dL)	298 ± 7	315 ± 2 (↑6%)	308 ± 6 (↑3%)	296 ± 5 (↓1%)
HDL-C (mg/dL)	40 ± 1	34 ± 4 (↓15%)	37 ± 1 (↓8%)	34 ± 3 (↓15%)
LDL-C (mg/dL)	83 ± 3	86 ± 5 (↑4%)	86 ± 7 (↑4%)	92 ± 3 (↑11%)
ALT (U/L)	30 ± 19	32 ± 15 (↑7%)	59 ± 28 (97↑%)	72 ± 33 (140↑%)
AST (U/L)	366 ± 174	254 ± 183 (↓31%)	140 ± 77 (↓62%)	416 ± 206 (↑14%)
Uric Acid (mg/dL)	4.1 ± 1	5.8 ± 3 (↑41%)	6.4 ± 2 (↑56%)	7.6 ± 3 (↑85%)
Creatinine (mg/dL)	0.2 ± 0.05	0.3 ± 0 (↑50%)	0.17 ± 0.09 (↓15%)	0.2 ± 0.1 (0%)
Total protein (mg/dL)	6 ± 0.81	6 ± 0 (0%)	6 ± 0 (0%)	6.3 ± 0.49 (↑5%)
Albumin (g/dL)	3.25 ± 0.5	3 ± 0 (↓8%)	3 ± 0 (↓8%)	3 ± 0.27 (↓8%)

This table is generated using mean ± standard deviation and percentage increase or decrease.

## Data Availability

No data set is linked with this study.

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
