# Peer review of "Histological and Microscopic Analysis of Fats in Heart, Liver Tissue, and Blood Parameters in Experimental Mice"

_genes, 2023, doi:10.3390/genes14020515_

Round 1
Reviewer 1 Report
Obesity and cardiovascular diseases (CD) are a common disorder and are alarmingly increasing in subcontinent, especially Pakistan. One of the main reasons for their increase is poor diet. This article empresses the role of diet on obesity and CD. This article can help clinicians and nutritionists to suggest optimal intake of these diets and may be help for patients with metabolic disorders.
However, there are some major issues for the research article.
Now, I will come to the other issues in the article.
1. The title does not show that the experiment was design based upon ethnic regimens. Its more elaborative than the work done or the field of the article.
2. The findings are very modest with known results, very low novelty.
3. Highly recommend rechecking grammatical issues, especially the articles.
4. Introduction is extensive and needless, plenty of information was not a part of the article. There should be more emphasis on the work done.
5. What is the mechanistic explanation of Table-1? Means, like glucose level is higher in HFD than C-ND but the body wight gain is different. Similarly, ALT levels are higher in HFD-DG and HFD-BG, was it good or bad?
6. What is the normal range of these chemicals, or which results were statistically significant?
7. First three paragraphs of the discussion should be a part of introduction.
Author Response
Manuscript ID: genes-2080594
Dear Editor-in-Chief and Reviewers of GENES
Please find attached a revised version of our manuscript, entitled “Histological and Microscopic analysis of fats in heart, liver tissue and blood parameters in experimental mice,” which we would like to submit for consideration for publication in GENES. This manuscript was originally submitted to your journal as assigned with genes-2080594, and then we have been recommended to revise the manuscript as per the reviewer’s comment and resubmit it. We thank the reviewers for thoroughly reviewing our manuscript, which greatly improved its quality. In the following pages are our point-by-point responses to each of the comments of the reviewers as well as your comments, and the changes made to the manuscript have been highlighted by red-colored text. We hope that the revisions in the manuscript and our accompanying responses will be sufficient to make our manuscript suitable for publication in GENES.
We shall look forward to hearing from you at your earliest convenience.
Sincerely Yours,
Dr. Imran Riaz Malik

Reviewer 2 Report
The paper is well-designed and structured. The research underlines the differences between various types of dietary fats on their cardiometabolic impact.
Comments:
The authors do not describe the statistical methods in the material and methods section. From de table 1, I supposed is media +/- SD, but it is not clear.
Need some English improvements.
The description of plaque formation is too extensive for the introduction, according to the purpose of the study. row 77-89
There is no section of conclusions.
Also, the abstract needs some improvements and better systematization, doesn`t underline conclusions.
Author Response

(The authors gave the same response as above.)

Round 2
Reviewer 1 Report
Please accept.
